# *Lactobacillus crispatus* S-layer proteins modulate innate immune response and inflammation in the lower female reproductive tract

Alexiane Decout [1] ✉, Ioannis Krasias[1], Lauren Roberts [1,2,3], Belen Gimeno Molina[1,2,4], Chloé Charenton[1], Daniel Brown Romero [1], Qiong Y. Tee[1], Julian R. Marchesi [2,3], Sherrianne Ng[1,2], Lynne Sykes [1,2,4], Phillip R. Bennett [1,2] & David A. MacIntyre [1,2]

*Lactobacillus* species dominance of the vaginal microbiome is a hallmark of vaginal health. Pathogen displacement of vaginal lactobacilli drives innate immune activation and mucosal barrier disruption, increasing the risks of STI acquisition and, in pregnancy, of preterm birth. We describe differential TLR mediated activation of the proinflammatory transcription factor NF-κB by vaginal pathogens and commensals. Vaginal *Lactobacillus* strains associated with optimal health selectively interact with anti-inflammatory innate immune receptors whereas species associated with suboptimal health including *L. iners* and *Gardnerella vaginalis* interact with both pro- and anti-inflammatory receptors. Anti-inflammatory action of *L. crispatus* is regulated by surface layer protein (SLPs)-mediated shielding of TLR ligands and selective interaction with the anti-inflammatory receptor DC-SIGN. Detection of SLPs within cervicovaginal fluid samples is associated with decreased concentrations of proinflammatory cytokines in *Lactobacillus crispatus*-dominated samples. These data offer mechanistic insights into how vaginal microbiota modulate host immune response and thus reproductive health and disease states.

Dominance of the vaginal microbiome by commensal *Lactobacillus* species, particularly *L. crispatus*, is widely associated with vaginal health. These species produce immunomodulatory and antimicrobial compounds that dampen inflammation and help to maintain the mucosal barrier thereby protecting against pathogen colonisation[1-5]. Conversely, *Lactobacillus*-depleted, high-diversity vaginal microbiota have been linked to mucosal inflammation[6,7] and adverse health outcomes such as bacterial vaginosis, Sexually Transmitted Infections (STI) acquisition,

miscarriage, and preterm birth[8-11]. Such non-optimal community compositions are often enriched with potentially pathogenic and BV-associated bacteria such as *Gardnerella vaginalis*, *Fannyhessea vaginae* and *Prevotella* species, which have been shown to stimulate pro-inflammatory cytokines production in several vaginal epithelial cell models in vitro[12-16]. In pregnancy, a shift from *Lactobacillus* dominance towards high-diversity vaginal microbiota associates with increased pro-inflammatory cytokine production and increased cervical vascularisation[9,17], which in women

[1]Imperial College Parturition Research Group, Institute of Reproductive and Developmental Biology, Department of Metabolism Digestion and Reproduction, Imperial College London, London, UK. [2]March of Dimes Prematurity Research Centre at Imperial College London, Hammersmith Hospital Campus, Du Cane Road, London, UK. [3]Division of Digestive Diseases, Department of Metabolism, Digestion, and Reproduction, St Mary's Hospital Campus, Imperial College London, London, UK. [4]The Parasol Foundation Centre for Women's Health and Cancer Research, London, UK. ✉e-mail: alexiane.decout@warwick.ac.uk

subsequently experiencing preterm birth, is characterised by excessive activation of the complement cascade[18].

The mechanisms determining innate immune response to vaginal microbiota remain poorly characterised however innate immune receptor-mediated activation of the transcription factor Nuclear Factor – kappa B (NF-κB) in both immune and epithelial cells is a central feature[10]. In addition to regulating gene transcription of pro-inflammatory mediators, NF-κB mediates expression of key tissue remodelling and pro-labour genes critical to human parturition[19–21]. Accordingly, untimely activation of innate immune receptors upstream of NF-κB has been associated with pregnancy complications[21]. During murine pregnancy, selective activation of TLR2 and TLR4-dependent signalling pathways in gestational tissues triggers preterm birth[20,22–24]. However, a TLR2-induced preterm birth phenotype is mediated by the co-receptor TLR1, and not TLR6, highlighting the complexity and specificity of innate immune recognition events in the reproductive tract[25]. Despite expressing the TLR2 ligands lipoteichoic acid and teichoic acids in their cell wall, commensal *Lactobacillus* spp. avoid activating pro-inflammatory pathways in the cervicovaginal niche[26]. In some species of lactobacilli S-layer proteins attached to cell wall carbohydrates by non-covalent interactions creating a lattice-like structure across the cell surface can modulate immune responses, including through interactions with anti-inflammatory receptors[27]. In the reproductive tract, these receptors including Siglec 10, which promotes sperm tolerance[28] and DC-SIGN (CD209), a receptor for HIV[29,30] play key roles determining host reproductive fitness. Interactions between microbiota and anti-inflammatory innate immune receptors in the vagina remains a largely unexplored area.

In this study we leverage a large collection of clinical vaginal bacterial isolates to describe the innate immune profile of major vaginal taxa. Our data indicates that *L. crispatus* interacts selectively with DC-SIGN while *L. iners* and BV-associated-bacteria strongly activate TLR2- and TLR4-dependent pro-inflammatory signalling. We show that the unique immunological properties of *L. crispatus* are mediated by their surface layer proteins (SLP), which mask TLR2 ligands from host recognition and mediate interaction with the anti-inflammatory receptor, DC-SIGN. Detection of SLPs in swabs collected during pregnancy associates with vaginal microbiota composition and may act as immunomodulators within the cervicovaginal niche.

## Results

### TLR2 and TLR4 are differentially activated by vaginal taxa associated with non-optimal health states

To understand the role played by vaginal microbiota in regulating the host immune environment, we investigated the capacity of patient-derived vaginal bacterial isolates to induce NF-κB and AP1 activation in HEK cells expressing human TLR2 (HEK TLR2) and TLR4 (HEK TLR4). Both the bacteria and bacterial culture supernatants were tested to evaluate direct activation and activation by secreted immunomodulatory compounds. Neither bacteria nor bacterial supernatants induced activation of the parental HEK-Null cells (Supplementary Fig. 1a, b). Isolates of *L. crispatus*, *L. jensenii* and *L. johnsonii* did not activate the TLR2 reporter cell line, except for *L. crispatus* 14M4 and 19N1 which induced low levels of activation (20-40% that of the positive control, Pam2CSK4) (Fig. 1a). *L. gasseri* 21M4 significantly activated the TLR2 reporter cell line, inducing 56% activation of the positive control ligand, while the vaginal isolates 19N2 and 18M1, and the commercial strain, DSM20077, did not activate TLR2 highlighting strain variability (Fig. 1a). Similarly, *L. vaginalis* 10N3, isolated from vaginal swabs, activated the TLR2 reporter cell line whereas the commercially sourced *L. vaginalis* DSM5837, did not (Fig. 1a). Culture supernatants containing secreted bacterial compounds of all isolates of *L. crispatus*, *L. gasseri*, *L. jensenii*, *L. johnsonii* and *L. vaginalis* did not activate the TLR2 reporter cell line with the exception of *L. vaginalis* 10N3 (Supplementary Fig. 1c). In contrast to these findings, clinical and commercial *L. iners* isolates and culture supernatants induced TLR2

activation to levels comparable to Pam2CSK4 (Fig. 1a and Supplementary Fig. 1c). Similarly, all *G. vaginalis* isolates and culture supernatants activated the TLR2 reporter cell line (Fig. 1a and Supplementary Fig. 1c).

Isolates of species associated with bacterial vaginosis and vaginal infections including *Sneathia vaginalis, Prevotella bivia, Mobiluncus mulieris, Escherichia coli, Fannyhessea (Atopobium) vaginae, Fusobacterium nucleatum, Klebsiella pneumoniae, Veillonella atypica* and *Finegoldia magna*[5,9,31,32] strongly activated the TLR2 reporter cell line to levels similar to Pam2CSK4 (Fig. 1a and Supplementary Fig. 1c). *Streptococcus agalactiae* supernatant, but not whole bacteria, induced low TLR2 activation (Fig. 1a and Supplementary Fig. 1c). The only bacteria found to activate the TLR4 reporter cell line were *Prevotella bivia, E. coli, K. pneumoniae, Fu. nucleatum* and *V. atypica* (Fig. 1b and Supplementary Fig. 1d).

TLR2-dependent immune activation by vaginal taxa was further investigated by examining IL-8 production in VK2 vaginal epithelial cells. While *L. crispatus, L. jensenii, L. johnsonii* and *L. vaginalis* isolates did not induce IL-8 production in VK2 cells, several isolates of *L. gasseri* and *L. iners* significantly induced TLR2-dependent IL-8 production (Fig. 1c). *G. vaginalis* induced IL-8 in a strain-dependent manner, whereas several BV-associated species including *S. vaginalis, P. bivia, E. coli, F. vaginae, Fu. nucleatum* and *Fi. magna* were also found to drive IL-8 production via TLR-2 (Fig. 1d).

### Vaginal taxa associated with sub-optimal health states activate TLR1/TLR2 pro-inflammatory signalling pathways

TLR2 forms heterodimers with either TLR1 or TLR6 to recognize triacylated and diacylated ligands respectively, and induce pro-inflammatory signalling[33,34]. To investigate TLR2 co-receptor dependency, we evaluated the ability of vaginal bacterial isolates previously shown to stimulate pro-inflammatory pathways, to activate the HEK TLR2 and VK2 cell lines in the presence of anti-TLR1 and TLR6 blocking antibodies. TLR2 signalling induced by isolates of *L. iners* (21N1 and 24C1) and *L. gasseri* (18M1, 19N2 and 21M4) was found to be strictly TLR6 dependent (Fig. 2a, b). Substantial isolate-to-isolate variability was observed among *G. vaginalis* vaginal isolates. While *G. vaginalis* 12N4 and 13C1 induced both TLR1- and TLR6-dependent TLR2 signalling, *G. vaginalis* 18C2 activated only TLR2/TLR6 signalling (Fig. 2a, b). Similarly, *S. vaginalis, M. mulieris, Fi. magna* and *V. atypica* activated only TLR6-dependent TLR2 signalling (Fig. 2a). *F. vaginae* and *Fu. nucleatum* were both found to activate TLR1- and TLR6- dependent signalling in HEK TLR2 and VK2 cell lines whereas *P. bivia* activated selectively TLR1-dependent signalling (Fig. 2a, b).

### *L. crispatus* selectively interacts with the anti-inflammatory receptor DC-SIGN

Anti-inflammatory innate immune receptors are known to play an important role in immune tolerance and immune evasion, including in the context of reproductive biology. We therefore examined the capacity of vaginal taxa to interact with three key anti-inflammatory receptors expressed in the female reproductive tract: Siglec 9[28], Siglec 10[28] and Dendritic Cell-Specific Intercellular adhesion molecule-3-Grabbing Non-integrin (DC-SIGN, CD209)[35,36]. Binding assays indicated that the majority of the lactobacilli isolates did not interact with Siglec 9 or Siglec 10, with the exception of *L. crispatus* UMB0085 and UMB0824, *L. gasseri* DSM20077 and *L. iners* DSM13335 binding weakly to Siglec 9, and *L. vaginalis* interacting with both Siglec 9 an Siglec 10 (Fig. 3a, b). In contrast, apart from *P. bivia*, all BV-associated bacteria displayed binding with Siglec 9 (Fig. 3a). *G. vaginalis* binding to Siglec 10 was isolate-dependent with binding observed for *G. vaginalis* UMB0540, UMB1414 and 18C2. No binding was seen between Siglec 10 and *G. vaginalis* UMB0233, UMB0358, UMB0775, DSM4944 and 12N4 (Fig. 3b). Interactions were also observed between Siglec10 and *S. vaginalis, E. coli, F. vaginae* and *St. agalactiae* (Fig. 3b).

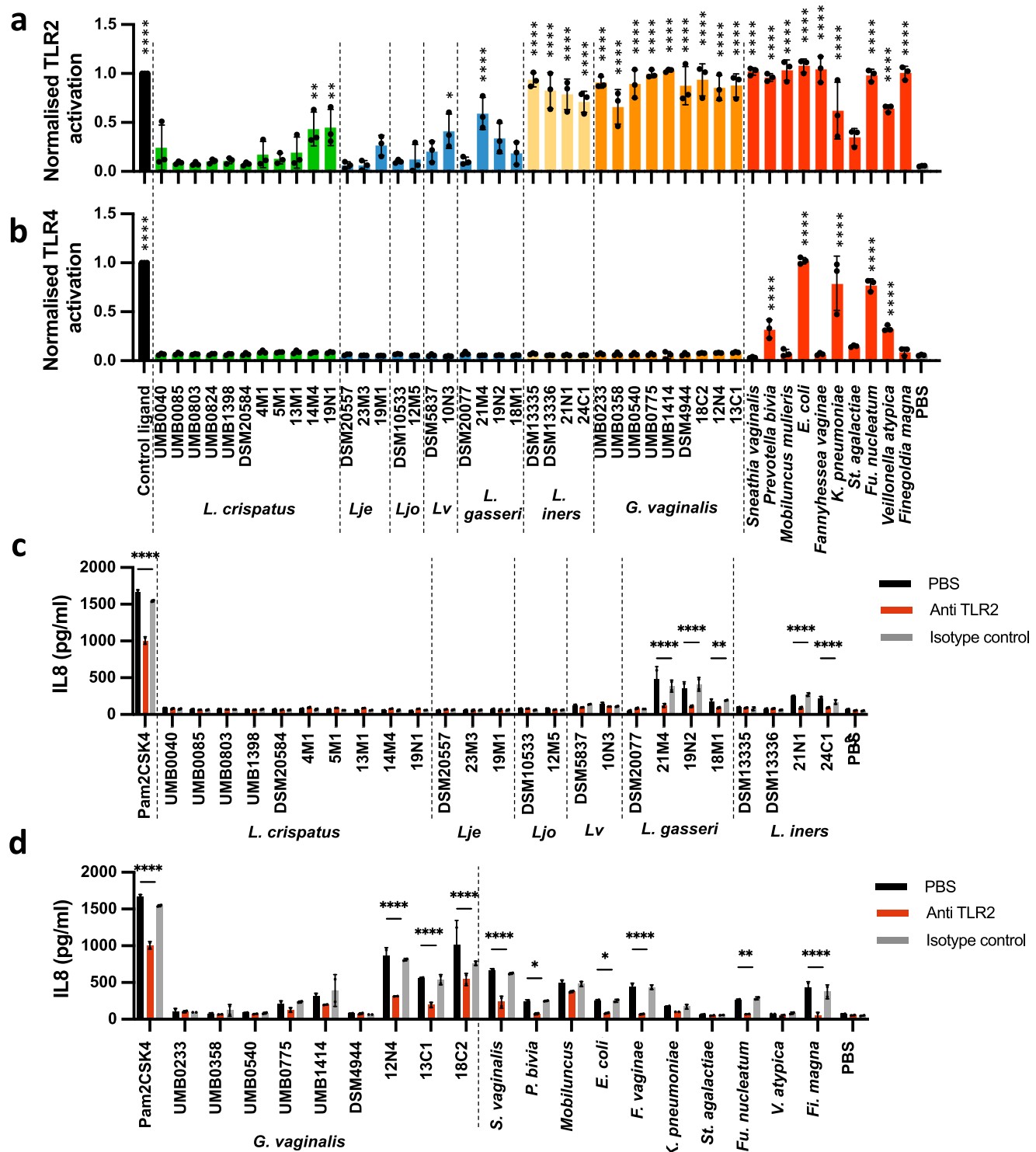

**Fig. 1 | Bacteria associated with preterm birth activate pro-inflammatory signalling pathways via TLR2 and TLR4.** HEK TLR2 (**a**), HEK TLR4 (**b**) and VK2 vaginal epithelial cells (**c**, **d**) were incubated with bacteria (MOI = 10) overnight. TLR2-dependence of IL-8 production on VK2 cells was investigated by pre-incubating the cells for 30 min with 1 μg/ml anti-TLR2 or isotype control antibodies before stimulation. Pam2CSK4 and LPS EK were used as control ligands for HEK TLR2 and HEK TLR4 respectively. Lje *Lactobacillus jensenii*, Ljo *Lactobacillus johnsonii*, Lv

*Limosilactobacillus vaginalis*. **a**, **b** Data are mean +/-SD (*n* = 3). One way ANOVA, Dunnett multiple comparison test against control (PBS) condition. **c**, **d** Data are mean +/−SD (representative of three independent experiments). Two-way ANOVA, Tuckey multiple comparison test between control (PBS) control and anti-TLR2. *$P < 0.05$, **$P < 0.01$, ***$P < 0.001$, ****$P < 0.001$. Source data are provided as a Source Data file.

Binding of vaginal taxa to DC-SIGN was highly variable between isolates. *L. crispatus* UMB0803 and UMB0824, *L. johnsonii* DSM10533 and *L. gasseri* DSM20077 binding to DC-SIGN was relatively high compared to other isolates (Fig. 3c). *G. vaginalis* isolates also

interacted with DC-SIGN, however, no binding was observed for *F. vaginae*, *K. pneumoniae* and *St. agalactiae*. DC-SIGN binding was also observed for *S. vaginalis*, *P. bivia*, *M. mulieris*, *E. coli*, *F. nucleatum*, *V. atypica* and *Fi. magna* (Fig. 3c). Consistent with previously described

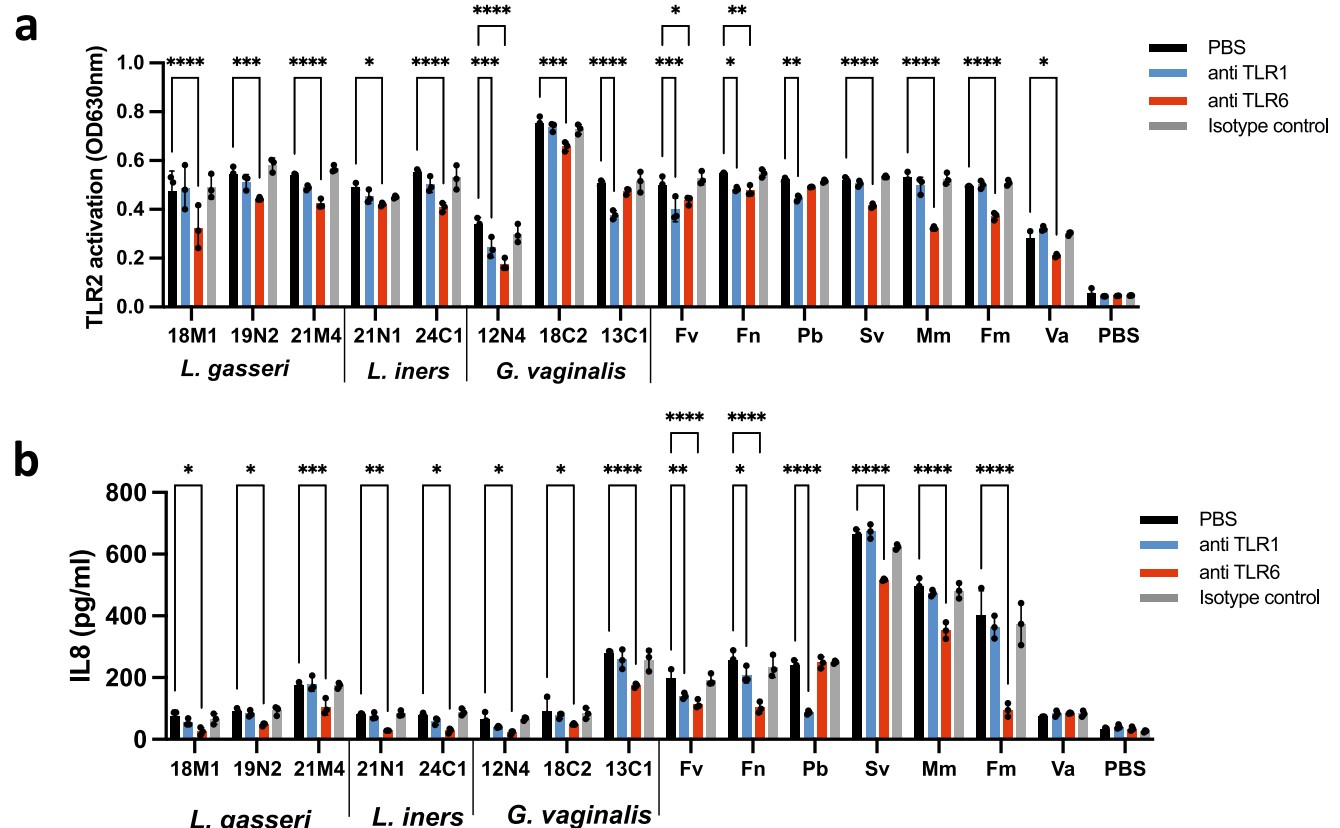

**Fig. 2 | TLR2 co-receptors utilisation by vaginal bacteria.** HEK TLR2 (**a**) and VK2 vaginal epithelial cells (**b**) were incubated with bacteria (MOI = 10) overnight. TLR1 and TLR6-dependence was investigated by pre-incubating the cells for 30 min with 1 µg/ml anti-TLR1, anti-TLR6 or mIgG1 isotype control antibodies before overnight stimulation with the bacteria at MOI = 10. Fv *Fannyhessea vaginae*, Fn *Fusobacterium nucleatum*, Pb *Prevotella bivia*, Sv *Sneathia vaginalis*, Mm *Mobiluncum mulieris*, Fm *Fingoldia magna*, Va *Veillonella atypica*. Data are mean +/−SD (*n* = 3 biological replicates). Two-way ANOVA, Tuckey multiple comparison test. *$p < 0.05$, **$p < 0.01$, ***$p < 0.005$, ****$p < 0001$. Source data are provided as a Source Data file.

DC-SIGN carbohydrate specificity[37–39], interactions between bacterial isolates and DC-SIGN were inhibited in presence of glucose, mannose and EDTA but not galactose (Supplementary Fig. 2a, b).

### S-layer proteins are selectively expressed by L. crispatus and shield TLR2 ligands

The cell wall of lactobacilli is known to contain TLR2 ligands such as lipoteichoic acids and lipoproteins[26]. S-layer proteins (SLPs), expressed by some lactobacilli, form a 2D crystalline array on the cell surface that mediates interactions with other cells including the potential shielding of cell wall constituents from detection[26,27]. Using an adapted lithium chloride-based extraction method[40], we next examined SLPs expression in *L. crispatus* and *L. iners* isolates. S-layer proteins within the expected relative molecular weight range could be isolated from all *L. crispatus* isolates[27] (Fig. 4a, Supplementary Fig. 3a, b). *L. crispatus* UMB0040, 4M1, 19N1 and 14M4 were found to predominately express the 60 kDa SLP1 (Fig. 4a) while *L. crispatus* 5M1, 13M1, UMB0085, UMB803, UMB824 and UMB1398 expressed the 45 kDa SLP2 (CsbA) (Fig. 4a, Supplementary Fig. 3a, b). Aggregation Promoting Factors (Afp), which share sequence homology with SLPs, have also been isolated from *L. johnsonii* and *L. gasseri* using lithium chloride extraction[41,42]. Proteins of corresponding molecular weights were detected in *L. johnsonii* DSM10533 (42 kDa) and *L. gasseri* DSM20077 (32 kDa) (Supplementary Fig. 3b). However, SLPs or Afp were not detected in any other *L. johnsonii*, *L. jensenii*, *L. gasseri* or *L. iners* vaginal isolates (Fig. 4a and Supplementary Fig. 3a).

As previously reported, *L. crispatus* isolates did not activate the HEK TLR2 reporter cell line. However, removal of SLPs from the cell surface using LiCl led to strong activation of the HEK TLR2 reporter cell line (Fig. 4b) and induction of IL-8 production by VK2 vaginal epithelial cells (Fig. 4c). Furthermore, activation of the TLR2 reporter cell line was shown to be TLR2- and TLR6-dependent, but not TLR1-dependent, for all *L. crispatus* isolates tested (Fig. 4d). SLPs purified from *L. crispatus* isolates did not activate the TLR2 reporter cell line (Supplementary Fig. 3c).

### SLPs from L. crispatus are ligands of DC-SIGN

We next investigated the capacity of SLPs isolated from different vaginal taxa to interact with the anti-inflammatory receptor, DC-SIGN. SLP1 purified from *L. crispatus* UMB0040, 4M1, 14M4 and 19N1 and SLP2 from *L. crispatus* DSM20584, UMB0824, UMB1398, 5M1 and 13M1 were shown to bind DC-SIGN by western blot (Fig. 5a and Supplementary Fig. 4a). Similarly, SLP isolated from *L. johnsonii* was also capable of binding DC-SIGN. Binding of DC-SIGN to *L. crispatus* SLPs was inhibited in the presence of glucose, mannose and EDTA but not galactose for all the strains tested (Fig. 5b and Supplementary Fig. 4c), indicating glycan-mediation of SLPs-DC-SIGN interactions. In contrast, lithium chloride-extracts from *L. iners* DSM13335 and DSM13336, *L. gasseri* DSM20077, *L. jensenii* DSM20557 and *L. vaginalis* DSM5487 did not bind DC-SIGN (Supplementary Fig. 4a, b).

### Detection of SLPs in human vaginal secretions and association with microbiota composition

Surface layer proteins from other bacterial species have been reported to be shed freely into the environment[43,44] or released via incorporation in extracellular vesicles[45]. We therefore investigated if

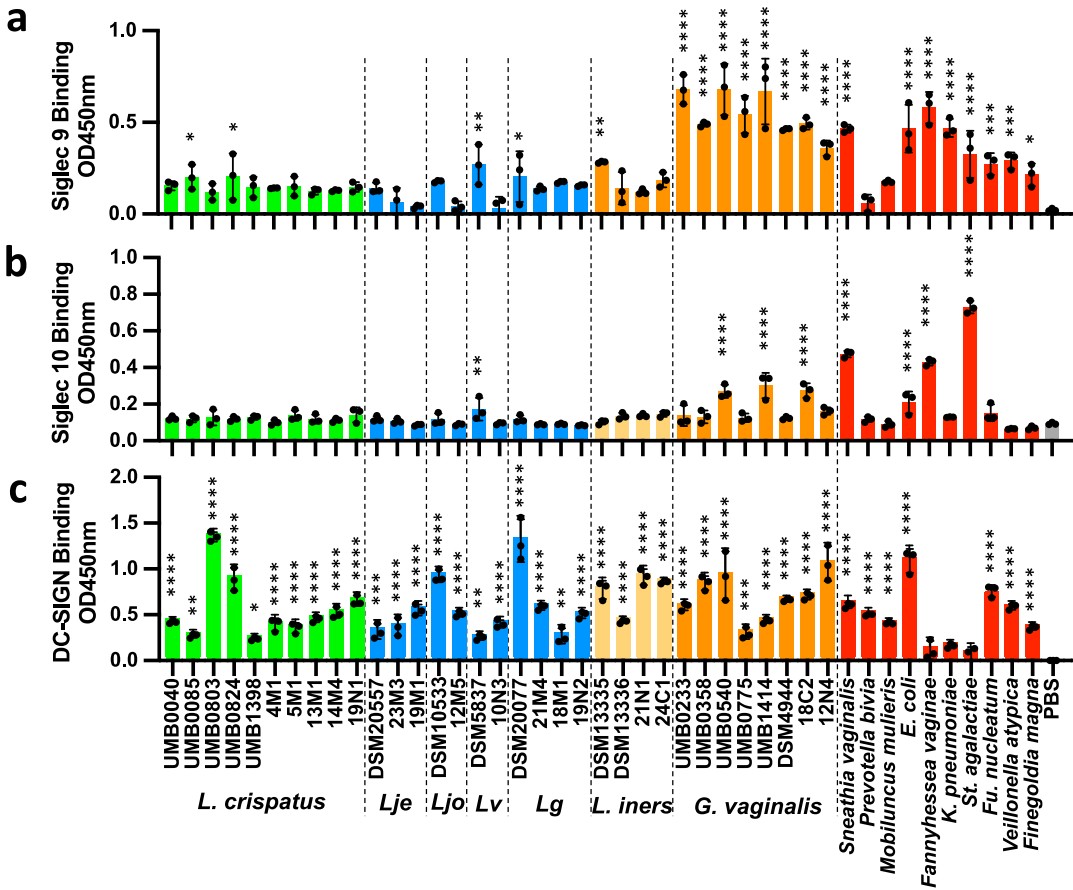

**Fig. 3 | Differential binding to inhibitory lectins by bacteria associated with term and preterm birth.** Bacteria (10⁶ per well) were coated in 96-well plates and tested for their capacity to bind Siglec 9-Fc (**a**), Siglec 10-Fc (**b**) and DC-SIGN-Fc (**c**). Lje *L. jensenii*, Ljo *L. johnsonii*, Lv *Limosilactobacillus vaginalis*, Lg *L. gasseri*. Data are mean +/−SD (*n* = 3 biological replicates). One way ANOVA, Dunnett multiple comparison test against control (PBS) condition. \**p* < 0.05, \*\**p* < 0.01, \*\*\**p* < 0.005, \*\*\*\**p* < 0.001. Source data are provided as a Source Data file.

SLPs could be detected in cervicovaginal fluids samples collected from a cohort of pregnant women with microbiota compositions dominated by *L. crispatus* (CST I, *n* = 26) or *L. iners* (CST III, *n* = 18) or *Lactobacillus*-depleted, high diversity compositions enriched for BV-associated bacteria (*n* = 12). SLPs were detected by western blot using an anti-SLP antibody (raised against *L. acidophilus* SLP and cross-reacting with SLPs from *L. crispatus*, *L. helveticus*, *L. amylovorus* and *L. gallinarum*) in 17 out of 26 CST I samples (78%), and 8 out of 18 CST III (44%) samples (Fig. 6a, b). Only 2 out of 11 CST IV samples (18%) were positive for SLPs, both of which contained *L. gasseri* or *L. crispatus* as a minor community member (Fig. 6a, Supplementary Fig. 5a and Supplementary Data 1). We next examined the interaction between SLP detection and immunomodulation in the cervico-vaginal fluids. No significant differences were observed in IL-1β, IL-6 and IL-8 production in presence or absence of SLP across all samples (Supplementary Fig. 5b–d). However, among *L. crispatus*-dominant samples, detection of SLPs in the cervico-vaginal fluids correlated with decreased IL-1β and IL-8 levels (Fig. 6c–e). Conversely, detection of SLPs correlated with increased IL-6 levels in *L. iners*-dominant samples (Fig. 6f–h). The Shannon index wasn't significantly different between the SLP positive and negative samples across all samples (Supplementary Fig. 5e) or among *L. crispatus* and *L. iners* dominated samples (Supplementary Fig. 5f, g). Finally, we investigated the capacity of the SLPs present in the cervicovaginal fluids to bind DC-SIGN in a subset of *L. crispatus* dominant SLP positive samples (*n* = 11). For all samples tested, SLPs were immunoprecipitated in presence of beads functionalised with recombinant DC-SIGN, but not mock beads (Fig. 6i and Supplementary Fig. 5h).

## Discussion

A loss of *Lactobacillus* species from the vaginal microbiota and a concurrent increase in bacterial diversity is associated with inflammation, predisposition to acquiring STIs and UTIs and during pregnancy, an increased risk of preterm birth[8–10,46]. However, host inflammatory response to vaginal taxa is highly heterogenous yet closely linked to pathophysiology[18,47,48]. For example, we have recently shown that during pregnancy women experiencing preterm birth are more likely to be carrying high diversity vaginal bacterial community compositions that are often, but not always, accompanied by high innate immune response[18]. By characterising and comparing innate immune signalling of major commensal and pathogenic bacteria isolated from human vaginal samples, the current study provides important insight into how vaginal bacteria differentially stimulate host-cell inflammatory response. Importantly, the majority of strains used in this study were isolated from the vagina of pregnant women and are representative of clinically and ecologically relevant strains, covering the most frequently identified vaginal bacterial taxa. Moreover, we report a mechanism by which *L. crispatus*, a commensal widely associated with optimal vaginal health, avoids stimulation of innate immune response via SLPs-shielding of TLR2/4 ligands and selective interacts with the anti-inflammatory receptor DC-SIGN.

The primary functions of the innate immune receptors, such as TLRs, Siglecs and DC-SIGN, are to trigger or modulate the host inflammatory response. Their relevance in bacterial sensing has been described in a broad range of clinical settings, from pulmonary infections to host-microbiota interactions[49–52]. In the context of women's reproductive health, TLR2 and TLR4 activation by purified ligands has

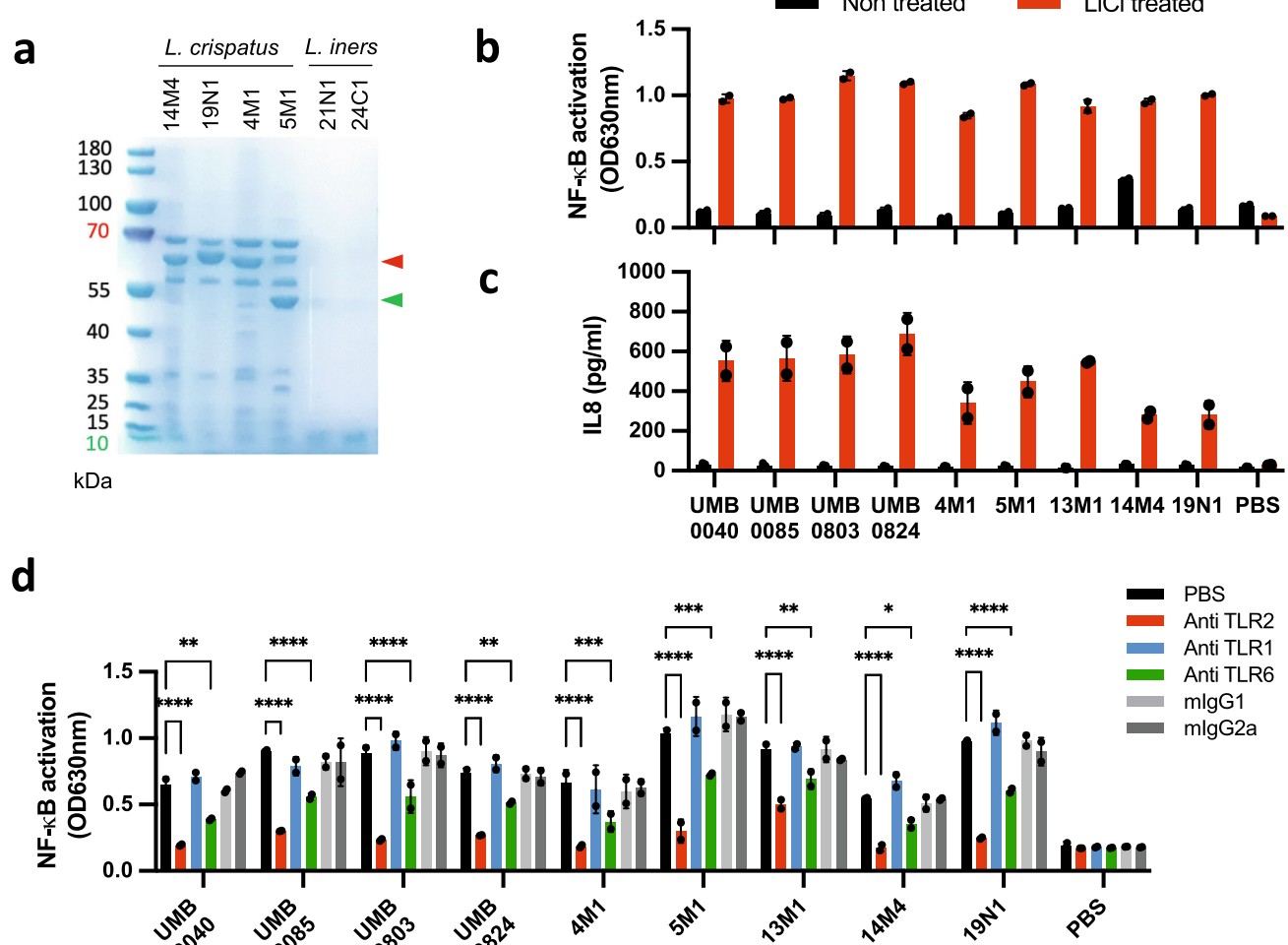

**Fig. 4 | SLPs mask TLR2 ligands of *L. crispatus* and prevent TLR2-dependent pro-inflammatory pathways activation. a** SDS-PAGE gel of crude SLPs extracted with LiCl 5 M from *L. crispatus* and *L. iners* isolates. 10 μg total proteins were loaded per lane. Representative of three independent experiments. **b** HEK-TLR2 cells were stimulated for 16 h with bacteria (MOI = 10) and NF-κB activation was determined by measuring alkaline phosphatase activity and reading O.D. at 630 nm. **c** VK2 vaginal epithelial cells were stimulated for 16 h and IL-8 release in the supernatant was quantified by ELISA. **d** HEK TLR2 cells were stimulated with lithium chloride-treated bacteria (MOI = 10) for 16 h and IL-8 release in the supernatant was quantified by ELISA. TLR1, TLR2 and TLR6 dependence was investigated by pre-incubating cells for 30 min at 37 °C with 1 μg/ml of anti-TLR1, anti-TLR2, anti TLR6 or mIgG1 and mIgG2b isotype control antibodies. **b**–**d** A representative figure of three independent experiments is shown (mean +/− SD). Two-way ANOVA, Tuckey multiple comparison test. *$p < 0.05$, **$p < 0.01$, ***$p < 0.005$, ****$p < 0,001$. Source data are provided as a Source Data file.

been shown to trigger preterm delivery in rodent and primate models, clearly indicating that untimely TLR-dependent activation of NF-κB signalling can be a driver of reproductive disorders[24,25]. Siglec 10 has been shown to promote sperm tolerance in the female reproductive tract[28]. DC-SIGN is also expressed in the female reproductive tract, and therefore is likely to contribute to the overall immune environment[30,36]. Using cell reporter systems, we found that clinical isolates representative of prevalent vaginal commensal and pathogenic species induce TLR signalling in a highly heterogenous manner. While *Lactobacillus* species, particularly those associated with optimal vaginal and reproductive health such as *L. crispatus* did not activate TLR2 or TLR4 pathways, species associated with sub-optimal vaginal health predominately activated TLR2 signalling in an isolate-dependent manner. This included *L. iners*, which often acts as a transitional coloniser of the cervicovaginal niche[53] and *G. vaginalis*, a species linked to BV and increased risk of preterm birth[9]. This variability appears to extend to vaginal epithelial cells where differential production of IL-8 was observed in response to exposure to differing vaginal taxa. These findings provide insight into recent in vivo studies which have observed high patient-to-patient variability in

inflammatory response to vaginal species[9,18]. These studies have been limited to the characterisation of vaginal microbiota composition to species taxonomy. Our data indicates that different strains of vaginal commensals and pathogens interact with TLR and anti-inflammatory receptors to differing degrees, which would partly account for the observed heterogeneity in host inflammatory response.

Isolate-to-isolate variability for *G. vaginalis* was also observed for binding of anti-inflammatory receptors, Siglec 9 and Siglec 10. Sub-species of *G. vaginalis* have been shown to differ in terms of virulence factors expressed, biofilm formation capacity and/or the presence of a polysaccharide capsule[25,42]. Such structural variations could underlie differing immune profiles as observed in this study, with exposure of anti-inflammatory molecular motifs acting as ligands for Siglec 9 and Siglec 10 thus modifying tolerance of potentially pathogenic species. Consistent with these findings, specific sub-species or strains of *G. vaginalis* have recently been implicated in increased risk of preterm birth[54,55]. We also observed variability of vaginal isolate binding to the anti-inflammatory receptor DC-SIGN. Despite this, strongest binding was observed for isolates of *L. crispatus*, the binding of which was shown to be mediated by specific glycan interactions. These findings

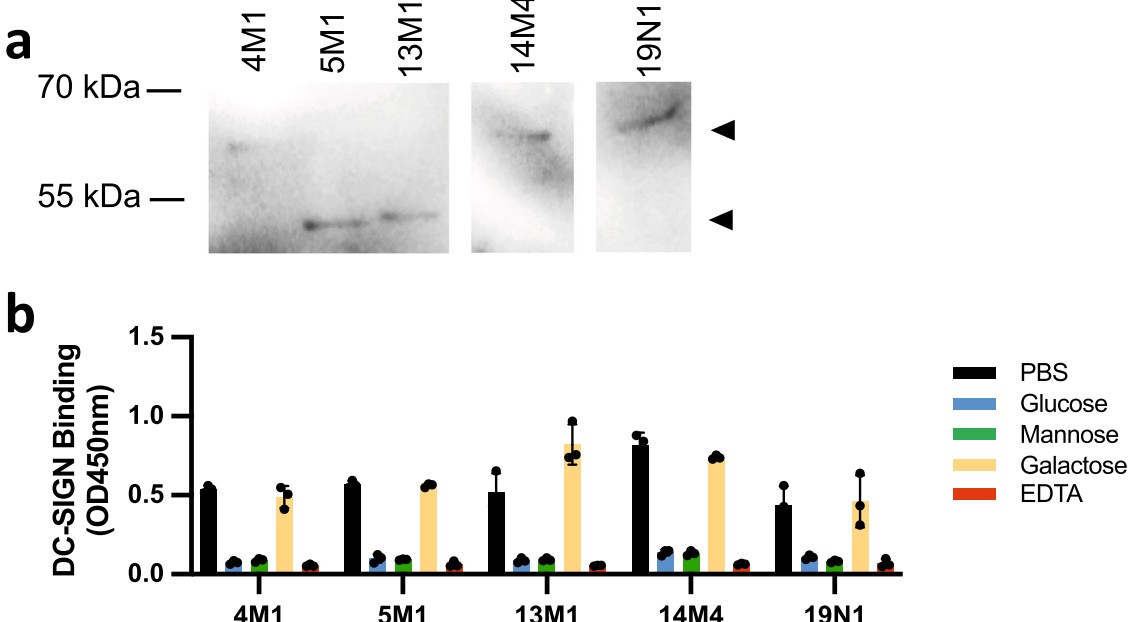

**Fig. 5 | SLPs isolated from *L. crispatus* vaginal strains are ligands of DC-SIGN.** **a** Western blotting results for DC-SIGN-Fc binding to purified SLPs isolated from vaginal L. crispatus strains (one representative of three replicates), **b** Purified SLPs isolated from vaginal *L. crispatus* strains (1 µg per well) were coated in 96-well plates and tested for their capacity to bind DC-SIGN-Fc. DC-SIGN-Fc (1 µg/ml) was pre-incubated or not with 20 mM EDTA or 40 mM glucose, mannose and galactose and allowed to react with the SLPs for 2 h at RT. Bound proteins were detected using a biotin-conjugated anti-IgG Fc specific antibody and avidin-HRP and reading O.D. at 450 nm. Data are mean +/−SD (*n* = 3 biological replicates). Source data are provided as a Source Data file.

are in agreement with variability observed in the predicted carbohydrate binding protein repertoire of vaginal pathogen and commensal species[56] and highlight the importance of examining the relationship between strain-level resolution of the vaginal microbiota, host inflammatory response and clinical outcomes.

Our findings also highlight SLPs as important mediators of host cell inflammatory response to *L. crispatus*. By shielding ligands contained within the cell wall of *L. crispatus*, SLPs were shown to prevent activation of pro-inflammatory signalling via TLR2 and moreover to be ligands for DC-SIGN. In our assays SLP1 and SLP2 seemed to play a redundant role in preventing TLR2 activation, however SLP2 has previously been shown to contain an amino-terminal domain that binds to type I and IV collagen and a carboxyl-terminal domains that modulates cell wall binding[57,58]. SLPs have been shown to dynamically regulate immune and inflammatory response to bacteria in other species via the modulation of bacterial cell wall permeability[59].

Our observation that SLPs are found not only on the surface of *L. crispatus* but also released in CVF indicates that secreted SLPs can broadly modulate the immunological environment of the vaginal niche. Interestingly, not all *L. crispatus* dominated CVFs had detectable levels of SLP. The mechanisms regulating SLP shedding by the lactobacilli are unknown but bacterial genomic and maternal environmental factors could possibly contribute to patient-to-patient variability. Detection of SLPs in *Lactobacillus crispatus* dominated samples clearly correlates with a decreased concentration of the pro-inflammatory cytokines IL-1β and IL-8, possibly indicative of an anti-inflammatory effect of the SLPs in *Lactobacillus crispatus* dominated samples. SLPs were also detected in some CST III samples, despite very low percentages of *L. crispatus* being detected. SLP production by *L. iners* has never been described and the molecular weight of the SLPs detected was identical to those observed in *L. crispatus* dominated samples. The presence of SLPs could reflect transitional colonisation by *L. crispatus*. In *Lactobacillus iners*-dominated samples, detection of SLPs seems to correlate with a slight increase in IL-6 concentration but IL-6 concentration was overall very low and close to the limit of detection. This

result should therefore be interpreted with caution. No effect on IL-1β and IL-8 levels was observed. Our method of detection didn't allow us to quantify the SLPs levels in the CVFs but the levels may be too low in *L. iners* dominated samples to prevent IL-1β and IL-8 production triggered by the more inflammatory strains.

In summary, our findings indicate that the vaginal lactobacilli associated with optimal health interact selectively with a very restricted subset of anti-inflammatory receptors through their Surface Layer Proteins, both in vitro and in cervico-vaginal fluids. We propose that these interactions shape and stabilize the maternal immune environment, preventing preterm parturition and other adverse health outcomes. Furthermore, bacterial strains activating TLR2 and TLR4 but also Siglec 9 and Siglec 10 are associated with bacterial vaginosis and adverse pregnancy outcome, indicating that some anti-inflammatory receptors may promote immune evasion and co-colonization by pathogenic strains rather than play a protective role. Overall, our data provide a rational ground to guide the selection and development of vaginal live biotherapeutic products with optimal innate immune properties, as well as host-targeted immunomodulatory therapeutics for the prevention and treatment of adverse women's health outcomes.

## Methods

### Patient recruitment and sample collection
Pregnant women at risk of preterm birth were recruited from preterm birth prevention clinics from Queen Charlotte's and Chelsea Hospital (QCCH) between June 2018 and June 2022. The study was performed under the Ethics approval REC 14/LO/0328 as part of the Vaginal Microbiome and Metabolome in Pregnancy (VMET 2) Research Study and approved by the NHS Health Research Authority (London -Stanmore Research Ethics Committee). All patients provided written informed consent to donate specimens.

Cervicovaginal Fluid (CVF) was first sampled using a BBL™ Culture Swab™ MaxV Liquid Amies swab (Becton, Dickinson and Company, Oxford, UK). Swabs were placed on ice immediately and stored at

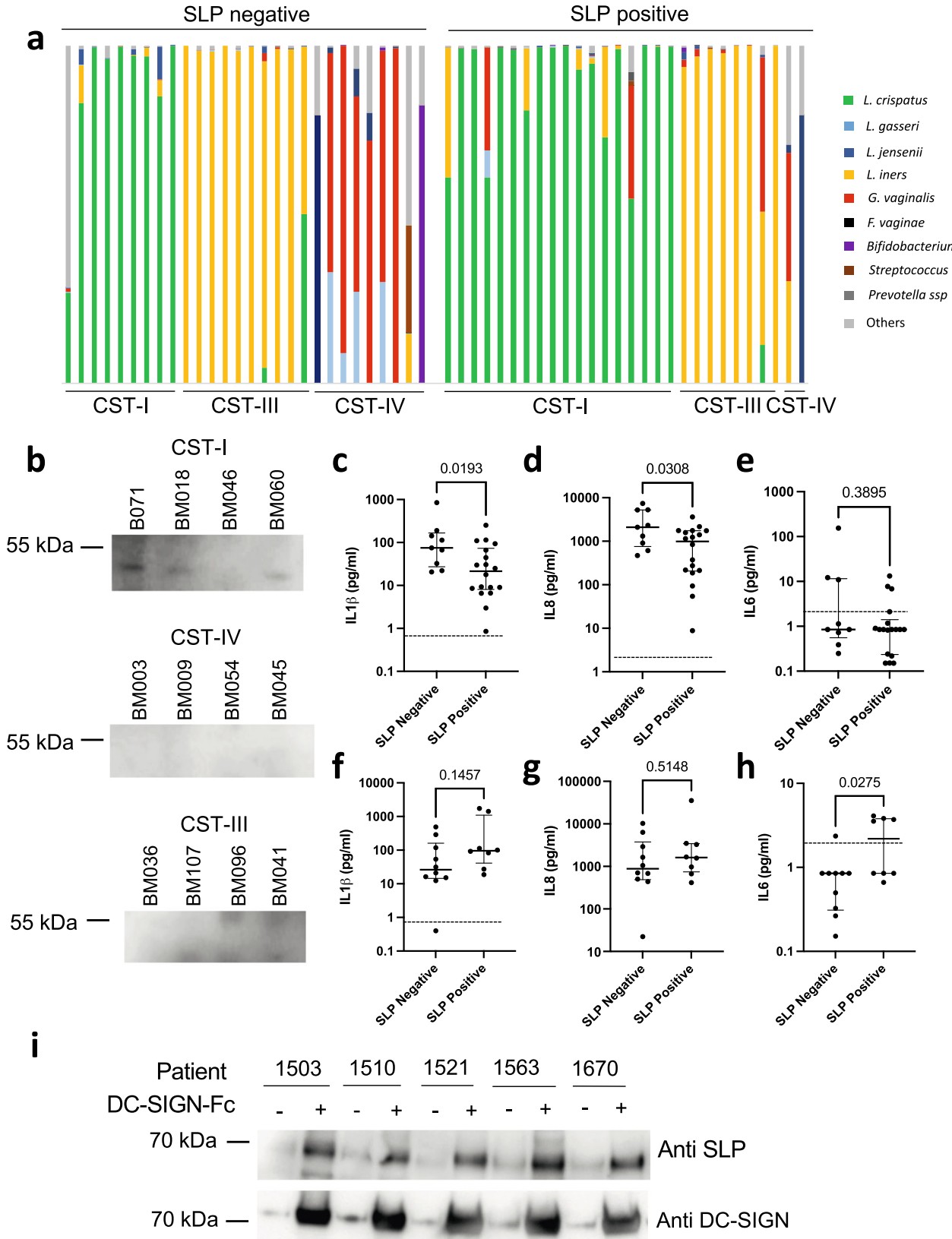

−80 °C until use. A soft cup (flexTM, menstrual disc) was then placed past the vaginal canal for 20 min, retrieved and the weight of the CVF collected and recorded. CVF from soft cups was mixed with 5 ml PBS per gram CVF and collected. Diluted CVFs were centrifuged at 4 °C for 10 min at 16,000 × *g* before the supernatant was collected and stored at −80 °C.

**Luminex immunoassays**

The BBL™ CultureSwab™ was thawed on ice. Supernatant was extracted from the sponge of the swab using a sterile syringe and pressure to release a volume of ~350 µl. This was centrifuged at 3000 × *g* for 10 min. IL-8 was measured on a single plex Human Premixed Analyte Kit (R&D Systems/Bio-Techne), following a tenfold dilution using

**Fig. 6 | SLPs are detected in cervico-vaginal fluids. a** Bacterial composition of the cervico-vaginal fluid samples determined by 16S rRNA gene sequencing, **b** Western blotting results for anti-Surface Layer Protein antibody binding to purified cervico-vaginal fluid samples. Representative of two technical replicates. Cervicovaginal concentrations of IL-1β (**c**), IL-8 (**d**), IL-6 (**e**) are shown from CST I samples ($n = 26$ women). Statistical analysis was performed using a one-sided Mann–Whitney test. Data are presented as median values and interquartile ranges (25th and 75th percentiles). Limit of detection is presented as dotted line. Cervicovaginal concentrations of IL-1β (**f**), IL-8 (**g**), IL-6 (**h**) are shown from CST III samples ($n = 18$ women). Statistical analysis was performed using a one-sided Mann–Whitney test. Data are presented as median values and interquartile ranges (25th and 75th percentiles). **i** Immunoprecipitation results for DC-SIGN-Fc binding to SLPs in CVF samples. Representative of three technical replicates. Source Data are provided as a Source Data File.

Calibrator Diluent RD6-52. The remaining cytokines, IL-1β and IL-6, were measured on a multiplex plate, Human Premixed Multi-Analyte Kit (R&D Systems/Bio-Techne) and no dilution was required. All samples were analysed on 96-well plates in duplicates, and data was acquired with Magpix™ and analysed with its software package control xPONENT® (version 4.3).

## Bacterial strains and culture conditions
Patient derived bacterial isolates used in this study are described in Supplementary Data 2 and 3[60]. Briefly, bacteria were cultured in their corresponding medium overnight at 37 °C. All the bacteria (except *E. coli* and *K. pneumoniae*) were grown anaerobically (10% $CO_2$, 10% hydrogen, 80% nitrogen; 70% humidity) and were washed in degassed HBSS. *E. coli* and *K. pneumoniae* were grown aerobically and washed with PBS. Aerobic liquid cultures were grown in a shaking incubator at 200 rpm. The bacteria were quantified as previously described[61]. The bacterial pellet was stored at −20 °C until further use.

## S-layer protein purification
Extraction of S-layer proteins was performed as previously described[40]. Briefly, bacterial pellets of 50 ml overnight cultures were washed with PBS and resuspended in LiCl 5 M at 4 °C for 15 min under stirring. The supernatants were harvested by centrifugating 10 min at $3000 \times g$ and dialyzed against water overnight at 4 °C in Snakeskin dialysis tubing cut off 10 kDa (68035, ThermoFisher Scientific). Precipitated S-layer proteins were recovered by centrifugation at $20{,}000 \times g$ for 20 min. The proteins were solubilized in LiCl 5 M and further purified by size exclusion chromatography using Sephacryl S200R (S200HR-250ML, Sigma Aldrich). The S-layer proteins were characterized by SDS-PAGE and stained with Coomassie Blue. A single band was obtained for all the strains.

## Binding of Fc-tagged lectins
Bacteria ($10^6$/well in isopropanol) or purified S-layer proteins (1 μg/well in isopropanol) were coated on 96-wells high binding plates (Sarstedt). DC-SIGN-Fc (10200-H01H-SIB-100ug, Sino Biological), Siglec-9-Fc (1139-SL-050, BioTechne) or Siglec-10-Fc (2130-SL-050, BioTechne) fusion proteins (1 μg/ml in PBS, 1 mM $CaCl_2$, 1% w/v BSA) were pre-complexed with anti-human Fc specific antibodies-HRP (A18817, ThermoFisher Scientific, 2:1 ratio lectin:antibody) and pre-incubated with 20 mM EDTA or 40 mM glucose, mannose or galactose (Tokyo Chemical Industry UK Ltd) when indicated. The recombinant lectins were allowed to bind to bacterial cells or SLPs for 2 h at RT (in 50 μl). Wells were washed once with PBS and the plates were incubated with a TMB (ES001-500ML, EMD). The reaction was stopped with HCl 1 M and the plates were read at 450 nm.

## TLR2 and TLR4 reporter cell lines experiments
The HEK-Blue hTLR2 and HEK-Blue hTLR4 (hkb-htlr2 and hkb-htlr4, InvivoGen), derivatives of HEK293 cells that stably express the human TLR2 and TLR4 genes respectively, along with a NF-κB-inducible reporter system (secreted alkaline phosphatase) were maintained in Dulbecco's modified Eagle's medium (DMEM, Sigma Aldrich) containing 10% v/v Fetal Bovine Serum (FBS, Sigma Aldrich), 2 mM L-glutamine, 100 U/ml penicillin and 100 μg/ml streptomycin (Sigma). HEK Null1 cells (hkb-null1, InvivoGen) were used as parental cell line.

Reporter cells ($5 \times 10^4$/well) were stimulated with bacteria (MOI = 10) or purified SLPs (from 10 μg/ml to 80 ng/ml) for 16 h, after which alkaline phosphatase activity was measured by mixing 10 μl of the culture supernatant and 90 μl of Quanti-Blue (InvivoGen) and reading O.D. at 630 nm.

## VK2 vaginal epithelial cells experiments
The VK2/E6E7 cells (CRL-2616, ATCC) were maintained in Keratinocyte Serum Free Medium (KSFM, Gibco) containing 100 U/ml penicillin and 100 μg/ml streptomycin (Sigma). The VK2/E6E7 cells ($3 \times 10^4$/well) were stimulated with bacteria (MOI = 10) or purified SLPs (from 10 μg/ml to 80 ng/ml). After 16 h, IL-8 was assayed in the culture supernatant using a commercially available kit (88-8086-77, ThermoFisher Scientific). To investigate TLR1, TLR2 and TLR6 dependence, VK2/E6E7 cells were pre-incubated for 30 min at 37 °C with 1 μg/ml of anti-hTLR1 (mabg-htlr1, clone H2G2, InvivoGen), anti-hTLR2 (MAB2616, clone # 383936, R&D Systems) and anti-hTLR6 (mabg-htlr6, clone C5C8, InvivoGen) antibody or mIgG1 (InvivoGen, clone T8E5, mabg1-ctrlm) and mIgG2b (14-4732-85, lot 2288614, clone eBMG2b, ThermoFisher Scientific) isotype controls.

## Immunoprecipitation
DC-SIGN ligands were immunoprecipitated using Protein A/G coupled beads (Bimake, B23201-BIT) functionalised with DC-SIGN-Fc (10200-H01H-SIB-100 ug, Sino Biological) according to manufacturer's instructions (10 ug rhDC-SIGN-Fc per 100ul beads). 10 ul of cervicovaginal fluids were added to 10 ul functionalised or mock beads in 200 ul TBS-T. The beads were incubated 1 h at RT and washed three times with TBS-T. The resin was boiled in Leammli 10 mM DTT before SDS−PAGE was performed.

## Western blot analysis
A total of 1 ug surface layer protein was added to Laemmli 10 mM DTT buffer followed by denaturing at 95 °C for 5 min. For soft cup samples, 10ul sample was used per lane. The proteins were separated by SDS−PAGE and transferred onto PVDF membranes. Blots were incubated with anti-Surface Layer Protein antibody (BS3797, polyclonal, Bioss), anti-DC-SIGN (R&D Systems, Clone # 120507, MAB161-100) and DC-SIGN-Fc (10200-H01H-SIB-100ug, Sino Biological) at 1 ug/ml in TBS-T 5% BSA. As secondary antibodies, anti-rabbit-IgG-HRP (1:2000) (Invitrogen, 31460), anti-mouse-IgG-HRP (1:2000) (Biotechne, NBP1-75130) or anti-human-IgG-HRP (1:2000) (ThermoFisher Scientific, A18817) were used respectively. ECL signal was recorded on the ChemiImager LAS4000 and data were analysed with ChemiImager LAS4000 software (version 1.3, GE Healthcare).

The CVF samples were considered as SLP-positive if a band was detected at 45 kDa or 60 kDa by western blot using the anti-Surface Layer Protein antibody (BS3797, polyclonal, Bioss). This antibody was raised against a synthetic peptide derived from *L. acidophilus* S-layer protein and has reported cross-reactivity against SLPs from *L. crispatus*, *L. helveticus*, *L. amylovorus* and *L. gallinarum*.

## DNA extraction and 16S rRNA gene sequencing
DNA extraction from the BBL™ CultureSwab™ was performed as previously described[62]. The V1-V2 hypervariable regions of bacterial 16S rRNA genes were amplified using a forward primer set (28f-/YM) mixed

to a 4:1:1:1 ratio of the following primers: 28F-Borrellia GAGTTTG ATCCTGGCTTAG, 28F- Chlorlex GAATTTGATCTTGGTTCAG, 28F-Bifido GGGTTCGATTCTGGCTCAG, 28F- YM GAGTTTGATCNTGGCTCAG; and a reverse primer that consisted of 388R GCTGCCTCCCGTAGGAGT. Sequencing was performed at Research and Testing Laboratories (RTL Genomics, Texas, USA). Cutadapt (version 2.8) was used for primer sequences trimming[63]. ASV (amplicon sequence variant) counts for each sample were computed using the QIIME2 bioinformatics pipeline (version2022.2.1)[64]. DADA2 (version 2022.2.0)[65] was utilised for denoising with forward and reverse read truncation legnths at 210 and 175 nucleotides respectively. Finally, the STIRRUPS database[66] was used for taxonomic classification of ASVs. At species level, samples were classed using the VALENCIA centroid classification algorithm into five community state types[67]. Shannon diversity was computed using the phyloseq package[68] (version 1.32.0) using R[69] (version 4.0.0).

## Statistical analysis

Prism software (Graphpad Software, version 10.1.1) was used to perform statistical tests and to generate graphs. Data are presented as mean ± s.d. and *P* values were calculated using Mann–Whitney test, one-way ANOVA and Dunnett's post-test or two-way ANOVA and Tukey's post-test as indicated in the figure legends.

## Reporting summary

Further information on research design is available in the Nature Portfolio Reporting Summary linked to this article.

## Data availability

The 16S rRNA gene sequencing data generated in this study have been deposited in the European Nucleotide Archive under accession code PRJEB83084. Source Data File includes the raw data used to create each figure. Source data are provided with this paper.

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

## Acknowledgements

We would like to thank all women who have participated in this study and members of the Women's Health Research Centre who facilitated and coordinated study recruitment and sample collection. *L. crispatus* UMB0040, UMB0085, UMB0803, UMB0824 and UMB01398, *G. vaginalis* UMB0233, UMB0358, UMB0540, UMB0775 and UMB1414 and *S. agalactiae* UMB0776 were kindly gifted by the Wolfe Lab. This work was funded by the March of Dimes European Preterm Birth Research Centre at Imperial College London and supported by the National Institute of Health Research (NIHR) Imperial Biomedical Research Centre (BRC). L.S. is supported by The Parasol Foundation Clinical Senior Lecturer scheme, and B.G.M. is supported by The Parasol Foundation Fellowship Scheme and The Rosetrees Trust. A.D. is funded by an Imperial College Research Fellowship.

## Author contributions

A.D., J.R.M., L.S., P.R.B. and D.A.M. designed research; A.D., I.K., B.G.M., C.C., D.B.R. and Q.Y.T. performed research; L.R. isolated and cultured the bacterial strains; S.N. processed the sequencing data; data processing, analysis, and interpretation was performed by A.D., I.K., L.R., B.G.M., C.C., D.B.R., Q.Y.T., J.R.M., L.S., P.R.B. and D.A.M.; and A.D. and D.A.M. wrote the first draft of the manuscript. All authors critically reviewed, read and approved the final manuscript.

## Competing interests

The authors declare no competing interests.
