## [Transparent Peer Review file · Nature Communications]

Lactobacillus crispatus S-layer proteins modulate innate immune response and inflammation in the lower female reproductive tract.

Corresponding Author: Dr Alexiane Decout

Version 0:

Reviewer comments:

Reviewer #1

(Remarks to the Author)

The merit of the paper by Decout et al is that it is one of the first attempts to reveal functional insights (gene-function relations) in *L. crispatus* as important vaginal microbiota member. Although the results on an S-layer protein of *L. crispatus* interacting with DC-SIGN and innate immune receptors are interesting, is only/merely based on in vitro work, without solid in vivo validation and does seem no major incremental finding in the field of (vaginal) lactobacilli and human microbiome. The idea at the end of the manuscript to look at S-lps in vaginal fluid is the most novel idea of the paper and could have been developed more to strengthen the manuscript.

In its present form, the paper seems incremental to work previously published on S-lp's of *L. crispatus* (indeed merely gut bacteria) and *L. acidophilus* (closely related to *crispatus*) interacting with DC-SIGN, both in terms of methodology (standard reporter cell lines) and number of isolates used. It seems a bit of a missed opportunity to not link the data and experiments better with vaginal microbiome/immune analysis from in vivo/clinical data.

In its present form, this papers also lacks a solid rationale for the selection of the strains use for the analysis. Are these the best model organisms? How can the authors document the relevance of their strains? which data do they have?

What does it mean if strains bind to specific TLRs and DC-SIGN? This is not studied (not in vitro or with in vivo data): the relevance seems based on literature, but potential pro-and anti-inflammatory functions of these receptors seem a bit overstated.

"Taxa associated with suboptimal" health also seem exaggerated: why would all members of a suboptimal community be recognized by one receptor and all members of an 'optimal community not? This seems too much simplification, especially since all experiments in this paper are done with single isolates and not communities.

Some data also seem to be rather quickly analyzed. For example, which statistics were used in figure 3?

To conclude, in its present form, I think this paper is more suited for a speciality journal.

Reviewer #2

(Remarks to the Author)

Summary: Decout et al report on a mechanistic characterization of vaginal bacteria, both pathogenic and commensal, and their interaction with various immune receptors. They demonstrate that S-layer proteins (SLPs) are produced by all clinical strains of *L. crispatus* but not *L. iners* and other lactobacilli, and that SLPs are directly responsible for binding an anti-inflammatory receptor, DC-SIGN, and preventing activation of TLR2 and IL-8 production by vaginal epithelial cells. SLPs were detected in the cervicovaginal fluids sampled from women with microbiota dominated by *L. crispatus* but not *L. iners* or with *Lactobacillus*-depleted communities. While the protective qualities of SLPs are well-known, this study importantly assesses a rich set of clinical strains and comprehensively characterizes immune response. The authors should make some

changes to improve the clarity and rigor of this manuscript, but overall the experiments presented are sound.

Comments:

1. In Figure 1A, it is difficult to understand the authors' threshold between "activated" and "not activated". For example, *L. jensenii* 19M1 is described as inducing low levels of activation while *L. crispatus* 13M1 appears to have similar activation but is not mentioned. A rationale for thresholding (e.g. percentage of positive control or significance) would be helpful. Additionally, significance markers should be included; *L. gasseri* is specifically noted in the text.
2. Similarly, in Figure 3, the threshold for "binding" is unclear. For example, in Figure 3A, some strains of *L. vaginalis* and *L. gasseri* appear to have higher binding of Siglec 9 compared to *M. mulieris*, but the text states that none of the lactobacilli interact with Siglec 9 and all BV-associated bacteria except *P. bivia* interact with Siglec 9.
3. In Figure 2, markers of non-significance make the graphs difficult to interpret and should be deleted.
4. The study would be strengthened with additional analysis of the cervicovaginal fluid, specifically to assess inflammation status such as levels of IL-6.

Version 1:

Reviewer comments:

Reviewer #2

(Remarks to the Author)

The authors have sufficiently addressed my comments.

Reviewer #3

(Remarks to the Author)

I've been asked to comment on the revisions performed to address Reviewer #1's concerns because Reviewer #1 was unavailable to review the revised manuscript. I have reviewed the manuscript in collaboration with an Early Career Researcher. In summary, we feel the authors have satisfactorily addressed a majority of Reviewer #1's comments, particularly those regarding the incorporation of in vivo/clinical data, strain selection, significance of TLR binding, and descriptions of statistical methods. We also agree with the authors that the paper does represent a significant step forward in the vaginal microbiome field, given that data from the gut microbiome field (i.e., regarding *L. acidophilus*) cannot simply be extrapolated to the vaginal microbiome. The clinical data are intriguing, but somewhat complex to interpret and do require some additional explanation and written discussion in the manuscript as requested below. But we believe a highly detailed evaluation of SLP functional activity within the human in vivo vaginal environment is well beyond the scope of what Reviewer 1 requested on the original review (and performing such a detailed evaluation could easily form its own, very interesting, follow up study). Nonetheless, some outstanding questions remain that should be addressed prior to publication, particularly in regards to the findings in the newly added Figure 6:

Major points:

- There needs to be more clarity on how clinical samples were defined as SLP positive or negative (ex. in figure 6A). If on an assay with an anti-SLP antibody, what is the specificity of this antibody and what were the criteria for positivity or negativity?
- The authors do not adequately address the somewhat surprising observation that there are *L. crispatus*-dominant samples both with and without SLPs. The authors should address this observation in their discussion -- does the absence of SLPs in some samples reflect strain-level differences or another phenomenon? Additional experimental data on this point would greatly strengthen the manuscript, but we believe this would be beyond the scope of the original request from Reviewer 1 and therefore it is not mandatory.
- It is not adequately addressed why there are SLPs present in CSTIII samples that have little or no *L. crispatus*, *L. jensenii*, or *L. gasseri*. Does this suggest that other lactobacilli, such as *L. iners*, can produce SLPs, but that these SLPs are functionally different from those produced by *L. crispatus*? Is there another potential explanation? Do these SLPs have the same molecular weight on western blot? If the SLPs are the same in CST III and CST I, how do the authors explain the different associations with cytokines?
- If the information is available, it would be helpful to know the BV status or vaginal microbiome composition of the source samples from which strains were isolated.

Minor points:

- Although it is probably beyond the scope of the current paper, it would be interesting to know if there are gene sequence-level differences in the SLPs present in strains that exhibit different phenotypes in vitro, and whether those genetic differences correlate with differences observed clinically in gene sequences from the metagenomes.
- There is a typo in line 294.

Reviewer #4

(Remarks to the Author)

Point to Point Reply to reviewers' comments

Reviewer #1 (Remarks to the Author):

The merit of the paper by Decout et al is that it is one of the first attempts to reveal functional insights (gene-function relations) in L. crispatus as important vaginal microbiota member. Although the results on an S-layer protein of L. crispatus interacting with DC-SIGN and innate immune receptors are interesting, is only/merely based on in vitro work, without solid in vivo validation and does seem no major incremental finding in the field of (vaginal) lactobacilli and human microbiome. The idea at the end of the manuscript to look at Slps in vaginal fluid is the most novel idea of the paper and could have been developed more to strengthen the manuscript.

In its present form, the paper seems incremental to work previously published on Slp's of L. crispatus (indeed merely gut bacteria) and L. acidophilus (closely related to crispatus) interacting with DC-SIGN, both in terms of methodology (standard reporter cell lines) and number of isolates used.

We thank the reviewer for their comments. However, we respectfully disagree that the study does not represent a major step forward in our understanding of how vaginal bacteria mediate host immune responses critically implicated in women's reproductive health. The vaginal microbiome, both compositionally and functionally, is vastly different from the gut microbiome. For example, high microbial diversity is considered a hallmark of health in the gut. In the vagina, it is associated with inflammation and increased risk of STI acquisition, cervical cancer and preterm birth. Hence, previous studies of gut bacterial strains cannot be directly applied to the vaginal niche. This point is further exemplified by the two species mentioned by the reviewer. Despite being closely related to *L. crispatus*, which often entirely dominates the vaginal microbiome, *L. acidophilus* is rarely found in the vaginal niche (1,2). The functional role of these species undoubtedly differs across human body niches.

To the best of our knowledge, our study is the first to provide such an extensive characterisation of innate immune response to vaginal bacterial pathogens and commensals and provides new mechanistic insight into how vaginal commensals might avoid exacerbation of inflammation in this niche. Our methodologies are robust and extend beyond the use reporter cell lines (See Figures 1C and D, 2B, 3, 4 A and C, 5 and 6).

It seems a bit of a missed opportunity to not link the data and experiments better with vaginal microbiome/immune analysis from in vivo/clinical data.

We do agree with the reviewers' suggestion that further clinical data would strengthen our manuscript. Considering these comments, we have increased the sample size of

our clinical patient cohort and undertaken new experiments to examine the relationship between the presence of Surface Layer Proteins in cervicovaginal fluid and host inflammation (Figure 6 and Extended Data Figure 5). These new findings show that in *L. crispatus*-dominated samples, SLP detection correlates with lower levels of IL-1b and IL-8, compared to samples where SLPs are not detected, indicating that secreted SLPs may contribute to the anti-inflammatory environment of the vaginal niche when dominated by *L. crispatus*.

In its present form, this papers also lacks a solid rationale for the selection of the strains use for the analysis. Are these the best model organisms? How can the authors document the relevance of their strains? which data do they have?

The vaginal microbiota can be classified into five major community state types based on the dominant bacterial taxa (2). Four of these are dominated by one species of *Lactobacillus*; CST I (*Lactobacillus crispatus*), CST II (*Lactobacillus gasseri*) CST III (*Lactobacillus iners*), and CST V (*Lactobacillus jensenii*). CST IV is characterised by low levels of *Lactobacillus* and high bacterial diversity with enrichment of potentially pathogenic species including e.g. *Gardnerella vaginalis*, *Fannyhessea vaginae*, *Prevotella bivia* or *Streptococcus agalactiae*. The majority of strains used in this study were isolated from the vagina of pregnant women, and are representative of clinically and ecologically relevant strains. We have now edited the discussion of the manuscript to provide this rationale for strain selection (See pg 8, line 335-338).

We have also updated Tables 1,2 and 3 in the Extended Data of the manuscript to include a more complete description of the bacterial strains used in the study. Extended Data Table 2 describes the commercially sourced bacterial strains as well as strains isolated from the urinary tract. Extended Data Table 3 describes the bacterial strains isolated from vaginal swabs alongside additional clinical information about the donors including blood group, ethnicity, age, BMI, parity and gravidity.

What does it mean if strains bind to specific TLRs and DC-SIGN? This is not studied (not in vitro or with in vivo data): the relevance seems based on literature, but potential pro-and anti-inflammatory functions of these receptors seem a bit overstated.

We thank the reviewer for highlighting the novelty around these findings. The primary functions of the TLRs and DC-SIGN are to trigger or modulate the host inflammatory response. Their relevance in bacterial sensing has been described in a broad range of clinical settings, from pulmonary infections to host-microbiota interactions. In the context of women's reproductive health, TLR2 and TLR4 activation by purified ligands has been shown to trigger preterm delivery in rodent and primate models, clearly indicating that untimely TLR-dependent activation of NF-kB signalling can be a driver of reproductive disorders. DC-SIGN is also expressed in the female reproductive tract,

and therefore is likely to contribute to the overall immune environment. This is the first study to report differential activation of these receptors by vaginal commensals and pathogens. We find the specific interaction between *L. crispatus* and DC-SIGN particularly intriguing. We have amended the discussion to more clearly describe the relevance of strain binding to these receptors (see Page 8, line 343-351).

"Taxa associated with suboptimal" health also seem exaggerated: why would all members of a suboptimal community be recognized by one receptor and all members of an 'optimal community not? This seems too much simplification, especially since all experiments in this paper are done with single isolates and not communities.

Association of specific vaginal taxa with suboptimal health is well described in the literature (3,4). A key feature of the vaginal microbiota is that bacterial taxa associated with optimal health are mutually exclusive, leading to domination of the vaginal niche by a single (*Lactobacillus*) bacterial species, except in the case of high diversity CST IV microbiota (5). The very distinct immune profile of *L. crispatus*, *L. gasseri* and *L. jensenii*, species widely associated with states of reproductive health, compared with bacterial species associated with suboptimal health is a key finding of our study. However, we also highlight that these species display a degree of strain-to-strain variability, which could have important clinical implications.

Some data also seem to be rather quickly analyzed. For example, which statistics were used in figure 3?

We thank the reviewer for pointing this out. We have now edited the text to clearly describe the statistics used for Figure 3 and have ensured that statistical significance compared to the negative control condition is appropriately described for Figures 1, 3 and Extended Data Figure 1.

Reviewer #2 (Remarks to the Author):

*Summary: Decout et al report on a mechanistic characterization of vaginal bacteria, both pathogenic and commensal, and their interaction with various immune receptors. They demonstrate that S-layer proteins (SLPs) are produced by all clinical strains of *L. crispatus* but not *L. iners* and other lactobacilli, and that SLPs are directly responsible for binding an anti-inflammatory receptor, DC-SIGN, and preventing activation of TLR2 and IL-8 production by vaginal epithelial cells. SLPs were detected in the cervicovaginal fluids sampled from women with microbiota dominated by *L. crispatus* but not *L. iners* or with *Lactobacillus*-depleted communities. While the protective qualities of SLPs are well-known, this study importantly assesses a rich set of clinical strains and comprehensively characterizes immune response. The authors*

should make some changes to improve the clarity and rigor of this manuscript, but overall the experiments presented are sound.

We thank the reviewer for the careful assessment of our article and appreciate their comments regarding the importance of comprehensively characterising the immune response of this rich set of clinically relevant vaginal bacterial isolates. We have amended our article in response to the reviewers' comments as detailed below.

Comments:

1. In Figure 1A, it is difficult to understand the authors' threshold between "activated" and "not activated". For example, L. jensenii 19M1 is described as inducing low levels of activation while L. crispatus 13M1 appears to have similar activation but is not mentioned. A rationale for thresholding (e.g. percentage of positive control or significance) would be helpful. Additionally, significance markers should be included; L. gasseri is specifically noted in the text.

We thank the reviewer for giving us the opportunity to clarify. Significance from non-stimulated condition (PBS) has now been added to Figure 1 and Extended Data Figure 1. The main text has also been modified accordingly to provide a clearer explanation of 'activated' and 'non activated'.

2. Similarly, in Figure 3, the threshold for "binding" is unclear. For example, in Figure 3A, some strains of L. vaginalis and L. gasseri appear to have higher binding of Siglec 9 compared to M. mulieris, but the text states that none of the lactobacilli interact with Siglec 9 and all BV-associated bacteria except P. bivia interact with Siglec 9.

We have now included a detailed description of the statistics used to analyse the binding data presented in Figure 3. Significance of binding relative to the negative control condition is now clearly presented and the main text has been modified accordingly (see Page 5, Line 162-169).

3. In Figure 2, markers of non-significance make the graphs difficult to interpret and should be deleted.

We thank the reviewer for pointing that out. The non significance markers have been removed, and the shape of the brackets modified to make the graphs easier to interpret.

4. The study would be strengthened with additional analysis of the cervicovaginal fluid, specifically to assess inflammation status such as levels of IL-6.

In consideration of these comments, we have now increased the size of our patient cohort and undertaken additional experiments to investigate the relationship

between the presence of Surface Layer Proteins in cervicovaginal fluid and maternal inflammation status. These new results show that in *Lactobacillus crispatus*-dominated samples, SLP detection is correlated with lower levels of IL-1b and IL-8 compared to samples where SLPs weren't detected. IL-6 levels were comparatively very low in our cohort.

References

- (1) May A. D. Antonio, Stephen E. Hawes, Sharon L. Hillier, The Identification of Vaginal *Lactobacillus* Species and the Demographic and Microbiologic Characteristics of Women Colonized by These Species, *The Journal of Infectious Diseases*, Volume 180, Issue 6, December 1999, Pages 1950–1956, <https://doi.org/10.1086/315109>
- (2) France, M.T., Ma, B., Gajer, P. *et al.* VALENCIA: a nearest centroid classification method for vaginal microbial communities based on composition. *Microbiome* **8**, 166 (2020). <https://doi.org/10.1186/s40168-020-00934-6>
- (3) Fettweis, J.M., Serrano, M.G., Brooks, J.P. *et al.* The vaginal microbiome and preterm birth. *Nature Medicine*, **25**, 1012–1021 (2019). <https://doi.org/10.1038/s41591-019-0450-2>
- (4) France, M., Alizadeh, M., Brown, S. *et al.* Towards a deeper understanding of the vaginal microbiota. *Nature Microbiology* **7**, 367–378 (2022). <https://doi.org/10.1038/s41564-022-01083-2>
- (5) Ravel, J. *et al.* Vaginal microbiome of reproductive-age women. *Proc. Natl. Acad. Sci.* **108** Suppl 1, 4680–4687 (2011). <https://doi.org/10.1073/pnas.1002611107>

Point to Point Reply to reviewers' comments

Reviewer #2 (Remarks to the Author):

The authors have sufficiently addressed my comments.

We thank Reviewer #2 for evaluating the revised manuscript.

Reviewer #3 (Remarks to the Author):

I've been asked to comment on the revisions performed to address Reviewer #1's concerns because Reviewer #1 was unavailable to review the revised manuscript. I have reviewed the manuscript in collaboration with an Early Career Researcher. In summary, we feel the authors have satisfactorily addressed a majority of Reviewer #1's comments, particularly those regarding the incorporation of in vivo/clinical data, strain selection, significance of TLR binding, and descriptions of statistical methods. We also agree with the authors that the paper does represent a significant step forward in the vaginal microbiome field, given that data from the gut microbiome field (i.e., regarding *L. acidophilus*) cannot simply be extrapolated to the vaginal microbiome. The clinical data are intriguing, but somewhat complex to interpret and do require some additional explanation and written discussion in the manuscript as requested below. But we believe a highly detailed evaluation of SLP functional activity within the human in vivo vaginal environment is well beyond the scope of what Reviewer 1 requested on the original review (and performing such a detailed evaluation could easily form its own, very interesting, follow up study). Nonetheless, some outstanding questions remain that should be addressed prior to publication, particularly in regards to the findings in the newly added Figure 6:

We thank Reviewer #3 and #4 for highlighting the novelty of our study.

Major points:

- There needs to be more clarity on how clinical samples were defined as SLP positive or negative (ex. in figure 6A). If on an assay with an anti-SLP antibody, what is the specificity of this antibody and what were the criteria for positivity or negativity?

SLP positivity was assessed on whole cervicovaginal fluids by western blot probed with a commercial anti-SLP antibody (Bioss, Ref BS-3797R). A sample was considered as SLP positive if a band was detected. The anti-SLP antibody was raised against a synthetic peptide derived from *L. acidophilus* S-layer protein and has reported cross-reactivity against SLPs from *L. crispatus*, *L. helveticus*, *L. amylovorus* and *L. gallinarum* (p7). Therefore, the SLPs detected could possibly come from other lactobacilli species, not exclusively from *L. crispatus*. We added a sentence in the main text and modified the “western blot analysis” part of the methods to clarify this point (p13-14).

- The authors do not adequately address the somewhat surprising observation that there are *L. crispatus*-dominant samples both with and without SLPs. The authors should address this observation in their discussion -- does the absence of SLPs in some samples reflect strain-level differences or another phenomenon? Additional experimental data on this point would greatly strengthen the manuscript, but we believe this would be beyond the scope of the original request from Reviewer 1 and therefore it is not mandatory.

It is indeed surprising that SLPs were undetected in some *L. crispatus* dominated samples. The bacterial diversity was the same in SLP-positive and SLP-negative samples (data added, see Extended Data Fig. 5e, f, g). Strain specific differences could underlie the ability of some isolates to shed their SLPs more efficiently than others. Maternal factors could also possibly influence the shedding capability of the bacterial strains. We added a section in the discussion to highlight this (p10).

- It is not adequately addressed why there are SLPs present in CSTIII samples that have little or no *L. crispatus*, *L. jensenii*, or *L. gasseri*. Does this suggest that other lactobacilli, such as *L. iners*, can produce SLPs, but that these SLPs are functionally different from those produced by *L. crispatus*? Is there another potential explanation? Do these SLPs have the

same molecular weight on western blot? If the SLPs are the same in CST III and CST I, how do the authors explain the different associations with cytokines?

There is no information in the literature regarding SLPs in *L. iners* and we didn't observe any SLP in the four *L. iners* isolates we studied. Furthermore, the SLPs detected in the CST III samples had the same molecular weight as *L. crispatus* SLPs. It is possible that the presence of SLPs in CST III samples reflects transitional colonisation by *L. crispatus*, *L. iners* being known to be a more unstable coloniser than other lactobacilli. Our method of detection for the SLPs didn't allow us to quantify the SLP levels in the CVFs. It is possible that the SLP levels are lower in CST III samples than in the CST I samples and are therefore insufficient to prevent inflammation. The discussion was modified to include these considerations.

- If the information is available, it would be helpful to know the BV status or vaginal microbiome composition of the source samples from which strains were isolated.

We agree that it would be very useful information. Unfortunately, microbiome composition wasn't collected when the strains were isolated and there is no sample left to analyse it. BV status wasn't recorded at the time of collection.

Minor points:

- Although it is probably beyond the scope of the current paper, it would be interesting to know if there are gene sequence-level differences in the SLPs present in strains that exhibit different phenotypes in vitro, and whether those genetic differences correlate with differences observed clinically in gene sequences from the metagenomes.

This is beyond the scope of our study but we agree that it is a very intriguing question. It is possible that the differences we observe between strains are both genetically encoded and environmentally regulated. Both mechanisms have been demonstrated as relevant in other bacterial species (Fagan and Fairweather, Nature Reviews Microbiology volume 12, pages 211–222 (2014)).

- There is a typo in line 294.

The typo was corrected.